# Therapeutic Effects of Inhibition of Sphingosine-1-Phosphate Signaling in HIF-2α Inhibitor-Resistant Clear Cell Renal Cell Carcinoma

**DOI:** 10.3390/cancers13194801

**Published:** 2021-09-25

**Authors:** Rouven Hoefflin, Sabine Harlander, Behnaz A. Abhari, Asin Peighambari, Mojca Adlesic, Philipp Seidel, Kyra Zodel, Stefan Haug, Burulca Göcmen, Yong Li, Bernd Lahrmann, Niels Grabe, Danijela Heide, Melanie Boerries, Anna Köttgen, Mathias Heikenwalder, Ian J. Frew

**Affiliations:** 1Department of Medicine I, Medical Center, Faculty of Medicine, University of Freiburg, 79106 Freiburg, Germany; rouven.hoefflin@uniklinik-freiburg.de (R.H.); behnaz.ahangarian@uniklinik-freiburg.de (B.A.A.); asin.peighambari@uniklinik-freiburg.de (A.P.); mojca.adlesic@uniklinik-freiburg.de (M.A.); philipp.seidel@uniklinik-freiburg.de (P.S.); kyra.zodel@uniklinik-freiburg.de (K.Z.); 2Institute of Physiology, University of Zurich, 8057 Zurich, Switzerland; sabine.harlander@uzh.ch; 3Zurich Center for Integrative Human Physiology, University of Zurich, 8006 Zurich, Switzerland; 4Institute of Genetic Epidemiology, Faculty of Medicine and Medical Center, University of Freiburg, 79106 Freiburg, Germany; stefan.haug@uniklinik-freiburg.de (S.H.); burulca.goecmen@uniklinik-freiburg.de (B.G.); yong.li@uniklinik-freiburg.de (Y.L.); anna.koettgen@uniklinik-freiburg.de (A.K.); 5Steinbeis Transfer Center for Medical Systems Biology, 69120 Heidelberg, Germany; bernd.lahrmann@stcmed.com (B.L.); niels.grabe@bioquant.uni-heidelberg.de (N.G.); 6Hamamatsu Tissue Imaging and Analysis Center (TIGA), BIOQUANT, University of Heidelberg, 69120 Heidelberg, Germany; 7National Center of Tumor Diseases, Medical Oncology, University Hospital Heidelberg, 69121 Heidelberg, Germany; 8Division of Chronic Inflammation and Cancer, German Cancer Research Center (DKFZ), 69120 Heidelberg, Germany; d.heide@dkfz-heidelberg.de (D.H.); m.heikenwaelder@dkfz-heidelberg.de (M.H.); 9Institute of Medical Bioinformatics and Systems Medicine, Medical Centre, Faculty of Medicine, University of Freiburg, 79106 Freiburg, Germany; melanie.boerries@uniklinik-freiburg.de; 10German Cancer Consortium (DKTK), Partner Site Freiburg, and German Cancer Research Center (DKFZ), 69120 Heidelberg, Germany; 11BIOSS Centre for Biological Signalling Studies, University of Freiburg, 79106 Freiburg, Germany

**Keywords:** HIF-inhibitors, sphingosine-pathway inhibition, HIF-2α resistance, clear cell renal cell carcinoma, tumor microenvironment

## Abstract

**Simple Summary:**

Clear cell renal cell carcinoma is a common malignancy that represents 80% of all kidney tumors. Most tumors harbor an inactivation of the *VHL* gene, leading to the accumulation of HIF-1α and HIF-2α. Promising clinical results of specific HIF-2α inhibitors will soon lead to new treatment options for advanced cancer patients, although primary and acquired resistance to these agents are common. We here show that Acriflavine, which inhibits both HIF-1α and HIF-2α, and Fingolimod (FTY720), which inhibits sphingosine-1-phosphate signaling, show therapeutic activities in several experimental ccRCC models that are resistant to HIF-2α-inhibitor treatment. Additionally, we show that specific HIF-2α-inhibition suppresses the tumor immune microenvironment, which will be important to consider for future combination studies with immune checkpoint inhibitors.

**Abstract:**

Specific inhibitors of HIF-2α have recently been approved for the treatment of ccRCC in VHL disease patients and have shown encouraging results in clinical trials for metastatic sporadic ccRCC. However, not all patients respond to therapy and pre-clinical and clinical studies indicate that intrinsic as well as acquired resistance mechanisms to HIF-2α inhibitors are likely to represent upcoming clinical challenges. It would be desirable to have additional therapeutic options for the treatment of HIF-2α inhibitor resistant ccRCCs. Here we investigated the effects on tumor growth and on the tumor microenvironment of three different direct and indirect HIF-α inhibitors, namely the HIF-2α-specific inhibitor PT2399, the dual HIF-1α/HIF-2α inhibitor Acriflavine, and the S1P signaling pathway inhibitor FTY720, in the autochthonous *Vhl/Trp53/Rb1* mutant ccRCC mouse model and validated these findings in human ccRCC cell culture models. We show that FTY720 and Acriflavine exhibit therapeutic activity in several different settings of HIF-2α inhibitor resistance. We also identify that HIF-2α inhibition strongly suppresses T cell activation in ccRCC. These findings suggest prioritization of sphingosine pathway inhibitors for clinical testing in ccRCC patients and also suggest that HIF-2α inhibitors may inhibit anti-tumor immunity and might therefore be contraindicated for combination therapies with immune checkpoint inhibitors.

## 1. Introduction

The defining genetic feature of the vast majority of clear cell renal cell carcinomas (ccRCC) is biallelic inactivation of the von Hippel–Lindau (*VHL*) tumor suppressor gene [1]. Loss of function of the pVHL protein prevents oxygen-dependent ubiquitin-mediated degradation of the HIF-1α and HIF-2α transcription factor subunits, leading to their constitutive stabilization and activation of transcription programs that contribute to numerous aspects of tumorigenesis. While these gene expression programs naturally show some overlap, it has become evident that a significant number of downstream targets are specific to either HIF-1α or HIF-2α and even mediate opposing cellular processes including metabolism, proliferation, and invasion [2]. While HIF-1α has been perceived in the past to act as tumor suppressor in advanced ccRCC [1], we recently demonstrated its crucial driver role in ccRCC formation [2], at least in the setting of the autochthonous ccRCC mouse model driven by deletion of *Vhl*, *Trp53,* and *Rb1*, arguing that HIF-1α inhibition should also be considered as a new ccRCC therapeutic concept in at least some settings.

Several lines of correlative and functional genetic evidence have ascribed a predominant oncogenic role to HIF-2α [3,4,5], which led to the development of a series of chemically related specific HIF-2α inhibitors PT2399, PT2385, and finally PT2977 (also known as MK-6482 and now as Belzutifan) which exhibits improved pharmacokinetics and is the molecule that has recently been clinically approved for the treatment of advanced tumors in VHL patients [6,7,8]. These drugs block the interaction of HIF-2α with the ARNT transcriptional co-activator and thereby inhibit HIF-2-dependent transcriptional activity [9]. HIF-2α inhibitors have shown excellent pre-clinical therapeutic activities in a subset of ccRCC cell lines and patient-derived xenograft models [6,7,10]. In heavily pretreated sporadic metastatic ccRCC patients, PT2385 and MK-6482 induced ORR (complete response plus partial response) of 14% [11] and 25% [12], respectively, while in the setting of localized ccRCC in familial VHL disease patients, MK-6482 induced confirmed RECIST response rates of 49%, leading to its FDA approval for several tumor types in VHL patients, including ccRCC. These studies demonstrate encouraging therapeutic efficacy of HIF-2α inhibition for ccRCC, and phase III clinical trials of Belzutifan are now ongoing (NCT04195750, NCT02974738). It is however also evident that resistance to HIF-2α inhibitors will become an important clinical challenge as some pre-clinical ccRCC cell line and patient-derived xenograft models are insensitive to HIF-2α inhibitors [6,10] and specific point mutations in HIF-2α or ARNT have been identified that abrogate the inhibitory effects of PT2385 and MK-6482 [9,13]. It would be desirable to have additional therapeutic options for the treatment of HIF-2α inhibitor resistant ccRCCs.

In ccRCC tumors that harbor a functional *HIF1A* gene, inhibition of HIF-1α, or of both HIF-1α and HIF-2α, might represent an alternative therapeutic strategy. We recently described an autochthonous mouse model of ccRCC based on the kidney epithelium-specific deletion of the *Vhl*, *Trp53,* and *Rb1* genes [14]. While the exact combination of triple *VHL*, *TP53,* and *RB1* mutations do not arise in human ccRCC, this model nonetheless reflects the frequent mutational inactivation of *VHL* combined with genetic dysregulation of cell cycle networks that commonly arise in human ccRCC involving multiple combinations of *TP53* mutation or copy loss, MDM2 and MDM4 mutation or copy gain, copy losses of CDKN1 and CDKN2 family genes, copy losses of RB family genes, and copy gains of CYCLIN family genes and CDK family genes [14]. ccRCC tumors arising in the mouse model also exhibit a very high degree of similarity at the transcriptomic [14] and proteomic level to human ccRCC [2], and also display sensitivity to Everolimus and Sunitinib [14], which are used clinically to treat ccRCC patients. We showed that the *Vhl* mutation is essential for the formation of ccRCC tumors in the *Trp53/Rb1* mutant background [14], reproducing one of the hallmark pathological features of human ccRCC and further showing genetically that tumor formation in this model is strongly dependent on HIF-1α but only weakly affected by genetic ablation of HIF-2α [2]. We showed that Acriflavine, an agent that acts to block the dimerization of both HIF-1α and HIF-2α with ARNT [15], exhibited therapeutic effects in some *Vhl/Trp53/Rb1* mutant ccRCCs that had been pre-treated with the tyrosine kinase inhibitor Sunitinib followed by the mTORC1 inhibitor Everolimus [14]. While Acriflavine does not possess ideal pharmacological properties for immediate clinical translation, it serves as a useful tool compound to pharmacologically inhibit both HIF-1α and HIF-2α in pre-clinical models. Indeed, Acriflavine exhibits anti-tumor activities in xenograft and autochthonous mouse models of several different types of tumors [15,16,17,18].

An alternative therapeutic strategy for combined HIF-1α and HIF-2α inhibition could potentially be to target upstream signaling. In this regard, the sphingosine kinase (SPHK)/sphingosine 1-phosphate (S1P)/sphingosine 1-phosphate receptor (S1PR) axis is of interest, since it acts as a master regulator of hypoxia by regulating HIF-1α and HIF-2α protein levels in several human cancer cell lines including *VHL*-deficient ccRCC [19,20]. SPHK1 is a lipid kinase that catalyzes the formation of S1P from the sphingolipid precursor sphingosine and acts as the rate limiting and therefore critical regulator of S1P-signaling [21]. In human ccRCC, SPHK1 is overexpressed and S1P serum levels are elevated in ccRCC patients, suggesting a pathway activation in this entity [22,23,24]. Moreover, VHL-loss (the defining feature of ccRCC) has recently been linked to SPHK1 accumulation through regulation of the Scm-like with four mbt domains 1 (SFMBT1) transcription factor [20]. SFMBT1 activity is essential for growth of ccRCC xenografts, and this effect appears to be at least partly mediated by its target gene *SPHK1* as genetic knockdown or pharmacological inhibition of SPHK1 inhibits ccRCC xenograft growth [20]. The soluble, bioactive phosphorylated lipid S1P is transported into the extracellular environment through transport proteins (ABCA1, ABCC1, SPNS2) where it acts as the ligand for a family of five S1P receptors (S1PR_1–5_) in an autocrine and paracrine manner. S1PRs are seven-transmembrane G protein-coupled receptors that stimulate a range of different signaling pathways in numerous cell types, including activation of the RAS and PI3K signaling cascades, and they play diverse roles in human tumors [21]. Of special interest is S1PR_1_ since, unlike S1PR_2–5_, it has been reported previously to be involved in the regulation of HIF-1α and HIF-2α expression in human ccRCC cell lines and its genetic inhibition blocked HIF-1α and HIF-2α protein accumulation [20]. Moreover, inhibition of S1P signaling by different strategies has been confirmed by multiple studies to inhibit the activity of HIF-1α and HIF-2α in ccRCC and other cancer cells [22,23,24,25] and both genetic and pharmacological inhibition of S1P signaling pathways induce anti-tumor activities in several ccRCC xenograft models [20,22,25,26,27,28]. In the context of potential further clinical development of S1P signaling inhibition for ccRCC therapy, it is important to note that FTY720 is a well-tolerated orally available drug that is FDA approved and chronically administered to treat multiple sclerosis patients. Its phosphorylated form FTY720-phosphate (FTY720-P) is an analogue of S1P and acts to downregulate S1PR_1,3–5_ family signaling with a preference of S1PR_1_ by acting as a superagonist and leading to receptor internalization [29]. In addition, the unphosphorylated form of FTY720 acts as an allosteric inhibitor of SPHK1, therefore inhibiting S1P-signaling irrespective of S1PRs [25,26].

In this study, we performed dual HIF-1α and HIF-2α inhibition using Acriflavine and S1P-signaling inhibition with FTY720 in an autochthonous ccRCC mouse model and human ccRCC cell lines, and showed evidence of a strong anti-cancer effect. Importantly, the efficacy was independent of a setting of specific HIF-2α-inhibitor resistance in vitro and in vivo, paving the way for future alternative ccRCC treatments.

## 2. Materials and Methods

### 2.1. Mice

Ksp1.3-CreER^T2^;*Vhl^fl/fl^*;*Trp53^fl/fl^*;*Rb1^fl/fl^* mice were previously described [14]. Tamoxifen (400 parts per million) was fed to 6-week-old mice for 2 weeks to induce gene deletion. Mouse crosses and phenotyping were conducted under the breeding license of the Laboratory Animal Services Center, University of Zurich and therapy studies were conducted under license ZH116/16 of the Canton of Zurich. 

### 2.2. Therapy Studies and Tumor Monitoring

Tumor growth monitoring was performed using µCT as previously described [14]. Tumor volumes were calculated every 7–14 days by measuring tumor diameters in all three dimensions (x, y and z-plane) using the formula for an ellipsoid (V = 4/3 × π × radius (x) × radius(y) × radius (z)). Treatment was initiated once tumor volume reached >5 mm^3^. FTY720 (MedChemExpress, Monmouth Junction, NJ, USA) was dissolved in 0.9% sodium chloride solution and 10 mg/kg bodyweight injected intraperitoneally every day. ACF (Sigma–Aldrich, St. Louis, MO, USA) was dissolved in PBS and 2 mg/kg bodyweight injected intraperitoneally twice daily. PT2399 (Peloton Therapeutics, Dallas, TX, USA) was dissolved in 25% ethanol and 75% polyethylene glycol 400 (Sigma–Aldrich) to create fresh stock solution every 3 days. Stock solution was then dissolved in a 3:2 ratio with water, containing 0.5% methylcellulose (Sigma–Aldrich) and 0.5% Tween 80 (Sigma–Aldrich). Fifty mg/kg bodyweight was administered via oral gavage twice per day.

### 2.3. Cell Culture, HIF-2α Mutagenesis and Retroviral Infections

Human ccRCC cell lines used in this study were 786-O, 769-P, UMRC-2, and A498, and they were cultured in either DMEM (Gibco, Thermo Fisher Scientific, Waltham, MA, USA) or RPMI (Gibco) media with 10% FCS (Sigma-Aldrich) and 1% Penicillin/Streptomycin (Gibco). A gatekeeper G323E mutation was introduced into the HA-HIF2α wt-pBabe-Puro plasmid [27] using the QuikChange II XL mutagenesis kit (Agilent Technologies, Santa Clara, CA, USA) and confirmed by Sanger sequencing. Retroviral infections of the wildtype or G323E mutated plasmid and sub-sequent puromycin selection of A-498 cells were performed as previously described [28].

### 2.4. Soft Agar Colony Forming Assay

Twelve-well plates were prepared by adding 1 ml of complete medium with 1% low melting agarose to create bottom layers. Cells (5 × 10^4^ per well) were suspended in complete medium with 0.4% agarose (Sigma–Aldrich) and added as the top layer. After solidification, 500 µL of medium was added to each well to prevent dehydration. Treatments with FTY720 (MedChemExpress), ACF (Sigma–Aldrich), PT2399 (Peloton Therapeutics), PT2385 (Merck KGaA, Darmstadt, Germany), and Siponimod (MedChemExpress) were initiated the next day and repeated 3 times per week over 4 weeks. Supernatant was aspirated and cells were stained with 0.1% iodonitrotetrazolium chloride (Sigma–Aldrich) overnight in a 37 °C incubator. Slides were scanned using a Zeiss Stereomicroscope (0.3× objective, 4× magnification) and colony quantification was performed using the fast cell count function of QuPath software (v0.2.3) [29].

### 2.5. Immunohistochemistry

Immunohistochemical stainings were performed as previously described [2,14]. Primary antibodies against the following epitopes were used at the following dilutions and antigen retrieval conditions: B220 (1:3000, BD Biosciences, Franklin Lakes, NJ, USA, 553084, Tris/EDTA), CA9 (1:2000, Invitrogen, Thermo Fisher Scientific, Waltham, MA, USA, PA1-16592, citrate), CD3 (1:250, Zytomed, Berlin, Germany, RBK024, citrate), CD4 (1:1000, eBioscience, Thermo Fisher Scientific, Waltham, MA, USA, 14-9766, citrate), CD8a (1:200, Invitrogen, 14-0808-82, citrate), CD69 (1:1000, Bioss, Woburn, MA, USA, bs-2499R, Tris/EDTA), F4/80 (1:250, Linaris Biologische Produkte, Dossenheim, Germany, T-2006, BOND Enzyme Pretreatment kit (Leica, Wetzlar, Germany, AR9551)), GLUT1 (1:200, Abcam, Cambridge, UK, ab14683, citrate), Ki-67 (1:4000, Abcam, ab15580, citrate), Ly-6G (1:800, BD, 551459, citrate), Perforin (1:100, Biorbyt, Cambridge, UK, orb312827, Tris/EDTA), and phospho-Thr37/Thr46-4E-BP1 (1:800, Cell Signaling Technologies, Danvers, MA, USA, 2855, citrate). All stained sections were scanned using a Nanozoomer Scansystem (Hamamatsu Photonics, Hamamatsu City, Japan). Digital quantification of immune cells expressing B220, CD3, CD4, CD8a, or CD68 was performed in duplicates using the VIS software suite (v4.5, Visiopharm, Hoersholm, Denmark) as previously described [2,30]. F4/80 positive cells were quantified using a positive pixel count and presented as the percentage positive pixel (%PP). Ki-67, Perforin, and CD69 were quantified using the positive cell detection function of QuPath software [29]. Ly-6G stainings were quantified manually by annotation of positively stained cells in the regions of interest.

### 2.6. Western Blotting

The following antibodies were used for western blotting: β-Actin (1:500, Sigma–Aldrich, A2228), Cyclin D1 (1:1000, Cell Signaling Technologies, E3P5S), NDRG1 (1:1000, Cell Signaling, #5196), Phospho-S6 Ribosomal Protein (Ser240/244) (1:1000, Cell Signaling, #2215), and Phospho-4E-BP1 (Thr37/46) (1:1000, Cell Signaling, 236B4). 

### 2.7. RNA-Sequencing, Gene-Set Enrichment Analysis and sc-RNA-Sequencing

Previously published sequencing data of wild type cortex and tumors from the Vhl^Δ/Δ^Trp53^Δ/Δ^Rb1^Δ/Δ^ mouse model was used for RNA-seq analyses [2,14]. Raw RNA sequencing data are available at GEO with identifier GSE150983 [https://www.ncbi.nlm.nih.gov/geo/query/acc.cgi?acc=GSE150983]. Publicly available raw RNA-seq data of the TCGA (The Cancer Genome Atlas) KIRC (kidney renal clear cell carcinoma) samples were downloaded from the NCI (National Cancer Institute) GDC (Genomic Data Commons) [31] using the R/Bioconductor package TCGABiolinks (version 2.18.0) [32] (project: TCGA-KIRC, data category: gene expression, data type: gene expression quantification, experimental strategy: RNA-seq, platform: Illumina; downloaded on 2 March 2021). From all samples, the 72 solid tissue normal samples and their matching 72 primary tumor samples from the same patient were extracted. Using the R/Bioconductor package DESeq2 (version 1.30.0) [33], the reads were normalized, lowly expressed genes were removed, and a variance stabilizing transformation was applied. Furthermore, log2 fold changes (log2FC) between the pooled tumor samples versus pooled normal samples were calculated. Using the R/Bioconductor package gage (generally applicable gene-set analysis) (version 2.40.1) [34], enrichment of signaling pathways was performed on the transformed counts from the ccRCC tumor samples from VpR mice [2] and the TCGA KIRC samples. The gene sets included a list of SFMBT1 targets [24] and all nine S1P pathway sets available from MSigDB [35]. The human gene identifiers from the MSigDB pathways were mapped on mouse homologues with the R/Bioconductor package GeneAnswers (R package version 2.28.0). *p*-values derived from Student’s *t* test and were adjusted via the Benjamini-Hochberg correction. Pathways were considered significant with an adjusted *p*-value < 0.05. For the visualization of the GSEA, enrichment plots were generated by ranking all genes for each dataset based on the log2FC. The GSEA_fgsea function from the R/Bioconductor package DOSE (disease ontology semantic and enrichment analysis) (version 3.16.0) [36] and the gseaplot2 function from the R/Bioconductor package enrichplot (version 1.10.2) were used to compute the statistics and generate the graphs. *p*-values were adjusted using the Benjamini–Hochberg correction.

The ccRCC scRNA-seq data from [37] were downloaded from https://singlecell.broadinstitute.org/single_cell/study/SCP1288/tumor-and-immune-reprogramming-during-immunotherapy-in-advanced-renal-cell-carcinoma (accessed on 15 June 2021). The tables with log-normalized counts, cluster assignments, and metadata were used to create a Seurat object with Seurat 4.0.4 [38]. Matrices with log-normalized counts for the indicated clusters and genes were exported with Seurat and violin plots and heatmaps were plotted with plotly 4.9.4.1. For heatmaps, the mean log-normalized counts per cluster were linearly rescaled to a range between 0 and 1 for each gene, with 0 being the minimum and 1 being the maximum counts per gene.

## 3. Results and Discussion

### 3.1. Evidence of Sphingosine Pathway Activation in Mouse and Human ccRCC

While we previously extensively characterized the transcriptional and proteomic consequences of HIF-1α and HIF-2α activation in the *Vhl/Trp53/Rb1* mutant mouse ccRCC model [2,14], the status of activation of transcription by SFMBT1 and of genes involved in the S1P signaling pathway was unknown. We therefore analyzed mRNA expression levels of *Sfmbt1* and of 16 genes that were recently identified to be specific SFMBT1 targets in human ccRCC [24] in normal mouse renal cortex and in mouse ccRCC tumors (Appendix A). The expression levels of nine of these genes were significantly upregulated in *Vhl/Trp53/Rb1* mutant tumors, consistent with the prediction of elevated activity of SFMBT1 as a consequence of deletion of the *Vhl* gene. Importantly, expression of the SFMBT1 target gene *Sphk1* encoding the rate-limiting enzyme of the S1P-pathway, was increased in tumors. Additionally, we used the SFMBT target genes to perform a general applicable gene-set enrichment (GAGE) that confirmed significant overexpression of the gene-set in *Vhl/Trp53/Rb1*-tumors vs. normal (padj = 0.001, Appendix A). The results were visualized using gene-set enrichment plots (Appendix A). We further examined the expression levels of additional genes involved in the S1P-signaling pathway and identified that *Abca1* and *Spsn2*, which encode plasma membrane S1P transporters, as well as *S1pr2*, encoding the S1PR_2_, were also upregulated in the mouse tumors (Appendix A), demonstrating that multiple components of the signaling pathway are highly expressed in the tumor model. This was further confirmed by performing GSEA of all nine S1P-signaling gene sets available at the Molecular Signatures Database (MSigDB). Four of the nine gene-sets revealed significant S1P-signaling activation based on GAGE-analysis (Appendix A). A similar analysis of S1P pathway genes in human ccRCC compared to normal kidney from the TCGA KIRC dataset revealed upregulation of *SPHK1*, as well as the *ABCA1* and *ABCC1* genes encoding S1P transporters and *S1PR1*, *S1PR4*, and *S1PR5* encoding S1PR receptors (Appendix A). Further, we repeated the GAGE analysis for SFMBT1 targets and the S1P-signaling gene sets with the KIRC TCGA data, confirming strong activation of these pathways in human ccRCC (Appendix A). In line with these transcriptional observations, a recent report showed significant SPHK1 protein expression in human ccRCC [24]. To further investigate protein expression levels involved in the S1P-pathway in ccRCC, we took advantage of the Human Protein Atlas platform [39]. Stainings were available for the S1P-transporters ABCA1, SPNS2, and ABCC1. ABCA1 and SPNS2 but not ABCC1 were highly expressed in the tumors (Appendix A) underlining their potential importance in facilitating S1P export from human ccRCC cells. This result is in line with the upregulated expression of *Abca1* and *Spns2* but not *Abcc1* in mouse ccRCC.

Thus, multiple components of S1P signaling are upregulated or strongly expressed in mouse and human ccRCC tumors, consistent with this pathway being a potential target of therapeutic intervention.

### 3.2. Acriflavine and FTY720 Show Antitumor Effects in HIF-2α-Resistant Mouse ccRCC

We induced gene deletion in *Ksp-CreER^T2^*, *Vhl^fl/fl^*, *Trp53^fl/fl^*, and *Rb1^fl/fl^* mice by feeding them at the age of 6 weeks for two weeks with food containing tamoxifen. Mice were imaged monthly by contrast-assisted μCT starting at 5 months after feeding to identify the timepoint of onset of tumor formation in individual animals. Tumors arose between 6 and 12 months after feeding and once they reached greater than 5 mm^3^ in volume the animals were randomly assigned to an untreated control group (*n* = 16 mice) or to groups treated for 14 days with a twice daily dose of PT2399 (*n* = 9), for 14 days with a twice daily dose of Acriflavine (ACF; *n*= 3) or for up to 30 days with a once daily dose of FTY720 (*n* = 7). The longer period of treatment with FTY720 was due to the encouraging therapeutic responses that were observed during the first two-week period of therapy (see below) to determine if these responses were sustained. All mice were monitored every 7–14 days using μCT to quantify tumor growth (Figure 1A).

To allow comparisons between treatments, growth rates were calculated over the first 14 days of treatment (Figure 1B). Consistent with our previous genetic demonstration that HIF-1α has a strong oncogenic function and HIF-2α possesses only a weak oncogenic function in this model [2], HIF-2α-specific inhibition with PT2399 did not affect tumor growth, whereas ACF and FTY720 reduced tumor growth rates. ACF had only a modest effect while FTY720 caused many tumors to either completely stop growing or to initially regress. In some cases, these regressions were short-lived, and tumors resumed growth after approximately 2 weeks in the presence of continued therapy. Tumors treated with PT2399 or FTY720 exhibited a reduced clear cell phenotype, while ACF treatment showed a non-significant trend towards reduced clear cell phenotype (Figure 1C,D). We have previously shown that the clear cell phenotype is genetically dependent on either HIF-1α or HIF-2α activity (Figure 1D) [2], providing a phenotypic readout of the activity of the drug treatments towards HIF-α factors. Interestingly, ACF and PT2399 treatments, but not FTY720 treatment, shifted the distribution towards higher nuclear grade (Figure 1E). Scoring of immunohistochemical stainings (Figure 2A) for GLUT-1 (Figure 2B) and CA9 (Figure 2C), the protein products of HIF-1α target genes [2], revealed that both ACF and FTY720 decreased abundance of these proteins, whereas PT2399 did not, indicative of efficacy of inhibition of HIF-α transcriptional activity by these drugs.

ACF and FTY720 treatments also decreased phospho-Thr37/Thr46-4E-BP1 staining (Figure 2D), a downstream readout of mTORC1 activity, a known driver of proliferation in ccRCC. None of the treatments caused strong reduction of the rate of positivity for the proliferation marker Ki-67 (Figure 2E), although FTY720 showed the strongest effect and induced an approximately 40% reduction in proliferation, consistent with the μCT imaging observations. It should be noted that the FTY720 data are biased towards showing higher Ki-67 positivity due to the fact that the treatment period was longer and several tumors had resumed growth in the presence of the drug. Given that the *Vhl/Trp53/Rb1* mutant ccRCC model displays exponential tumor growth [2], which does not reflect the more indolent, slow-growing nature of the majority of human ccRCC tumors, it can be speculated that the hurdle to achieving therapeutic effects is higher in the aggressively growing mouse model than in the human setting. It is also likely that therapeutic resistance, as observed for FTY720, will emerge more rapidly in the mouse model. We therefore concluded that FTY720 and to a lesser extent ACF show encouraging therapeutic effects in a mouse ccRCC model that is genetically independent of HIF-2α and pharmacologically resistant to HIF-2α inhibition.

### 3.3. Impact of Pharmacologic HIF-Inhibition on the Tumor Immune Microenvironment

We have recently shown that tumor cell-specific deletion of *Hif1a* or *Hif2a* differently affect the tumor microenvironment in mouse ccRCC, with genetic deletion of *Hif2a* leading to a stronger anti-tumor response characterized by increased numbers of active CD8^+^ T cells [2]. Pharmacological inhibition of HIF-1α and/or HIF-2α may therefore represent a strategy to alter the tumor microenvironment to promote anti-tumor immunity. However, a potential caveat is that the HIF-α transcription factors have pleiotropic functions in many different immune cells and these activities will be simultaneously inhibited along with HIF-α activities in tumor cells. To investigate this issue, we characterized whether PT2399, ACF, and FTY720 treatments affected the composition of the tumor immune microenvironment by immunohistochemically quantifying (Figure 3A–I) the numbers of immune cells expressing markers of T cells (CD3), helper T cells (CD4), effector T cells (CD8), early activated T cells and NK cells (CD69), cytotoxic T cells and NK cells (Perforin), B cells (B220), monocytes and macrophages (CD68), differentiated macrophages (F4/80), and granulocytes and neutrophils (Ly6G) in normal and tumor tissue.

Compared to untreated tumors, ACF-treated tumors had fewer CD69 and Perforin positive cells, indicative of suppressed T cell activation, as well as reduced numbers of granulocytes/neutrophils. PT2399 showed strong inhibitory effects on the tumor immune microenvironment, greatly decreasing the number of CD8, CD69, and Perforin positive cells, as well as decreasing the number of B220 and F4/80 positive cells. Despite Fingolimod (FTY720) being used clinically as a CD4-specific immunosuppressive agent in the setting of therapy for multiple sclerosis, FTY720 did not alter lymphocyte numbers but caused a reduction in the expression of markers of activated T cells, albeit to a lesser extent than PT2399 or ACF, as well as a strong reduction of numbers of F4/80 positive macrophages. These studies suggest that treatment of ccRCC patients with HIF-2α inhibitors may have the downside of inhibiting anti-tumor immunity. Given that HIF-2α inhibitors show encouraging results in clinical trials, in particular in the setting of treatment of VHL disease patients where long-term drug therapy is envisaged, it would be important to determine clinically whether this therapeutic intervention might also have specific inhibitory effects on anti-tumor immunity or cause more general immunosuppressive effects. Our pharmacological findings are consistent with a series of studies showing that HIF-α transcription factors are important for correct cytotoxic T cell differentiation and function in anti-tumor immunity [40,41,42,43,44] and it was recently shown that activated CD8^+^ liver T cells express high levels of HIF-2α and PT2385 suppressed the effector functions and survival of these cells [45]. While a phase I clinical trial of PT2385 together with nivolumab has been carried out [46], these findings also suggest that the combination of HIF-2α inhibition with immune checkpoint blockade is unlikely to be a viable therapeutic strategy. On the other hand, FTY720 therapy demonstrated anti-tumor effects in this model with more modest effects on the immune microenvironment, suggesting that this therapy should be prioritized for clinical trials. These studies more generally highlight that it will be important to analyze the effects of HIF-α inhibitory therapeutic interventions on the immune system in greater detail in patients.

### 3.4. Acriflavine and FTY720 Inhibit Growth of HIF-2α-Resistant Human ccRCC Cell Lines

We next sought to investigate whether FTY720 and ACF also show efficacy in human ccRCC cell models that are resistant to HIF-2α inhibitors. It was recently identified that human ccRCC cell lines differently require HIF-2α to grow as colonies in soft agar [6]. We therefore used the soft agar colony assay as a functional HIF-2α-dependent assay in UMRC-2 and 769-P cells in which sgRNA-mediated HIF-2α knockout or pharmacological inhibition of HIF-2α does not inhibit colony formation and in A498 cells which are genetically and pharmacologically sensitive to loss/inhibition of HIF-2α [6]. Colony-forming assays (Figure 4A) were conducted using all three cell lines in the presence of different doses of two different HIF-2α inhibitors (PT2385 and PT2399), ACF, or FTY720, and colony numbers were quantified (Figure 4B–D).

The absence of inhibition of colony formation in UMRC-2 and 769-P cells by PT2385 or PT2399 treatment, as well as the sensitivity of A498 cells to these agents, validates our assay conditions against the previous findings [6] (Figure 4B–D). Importantly, soft agar colony formation in all three cell lines was inhibited by ACF and by FTY720 in a dose-dependent manner (Figure 4B–D). 

To further confirm the antitumor efficacy of ACF and FTY720 in an HIF-2α-inhibitor-resistant setting, we created A498 cell lines with retrovirally-mediated ectopic expression of either wild-type HIF-2α (*HIF2A* WT) or a mutated G323E HIF-2α (*HIF2A* G323E). The HIF-2α G323E substitution interferes with drug binding [47] and ectopic expression of mutated HIF-2α is sufficient to prevent drug-induced dissociation of HIF-2 complexes [10] as previously confirmed in a human ccRCC metastasis [13]. HIF-2α-inhibition with PT2399 showed partial and dose-dependent treatment rescue in *HIF2A* G323E cells as indicated by the downstream targets NDRG1 and CYCLIN D1 (Figure 4E) [6]. To confirm this rescue, we repeated the colony formation assay using uninfected A498 cells together with *HIF2A* WT and *HIF2A* G323E cells (Figure 4F). As expected, uninfected A498 and *HIF2A* WT cells showed significant and dose dependent reduction in colony formation upon treatment with PT2399. This effect was partially but significantly rescued in *HIF2A* G323E cells in a dose-dependent manner (Figure 4G), while the efficacy of ACF and FTY720 remained unaffected (Figure 5A).

We conclude that ACF and FTY720 show anti-proliferative activities in cell lines irrespective of their sensitivity to HIF-2α inhibition.

### 3.5. Multiple S1P Receptors Contribute to ccRCC Tumor Cell Proliferation 

FTY720 is a functional pan-S1P receptor inhibitor acting on four of the five receptors (S1PR_1,3–5_). In contrast, Siponimod (Sipo) (also known as Mayzent) is a new functional S1P-receptor inhibitor with narrower target selectivity (S1PR_1,5_) and was recently approved for the treatment of relapsing forms of multiple sclerosis after showing efficacy in a phase III clinical study [48]. To begin to prioritize these two S1PR inhibitors for future clinical testing for ccRCC, we next compared the anti-proliferative efficacy of Sipo and FTY720 in colony formation assays. Interestingly, Sipo did not show any antiproliferative capacity in the three human ccRCC cell lines investigated (Figure 5A–C).

Part of the explanation for this result likely comes from the expression patterns of the S1PR family members in ccRCC. We conducted RNA-sequencing of five human ccRCC cell lines (769-P, A498, 786-O, SLR22, RCC4) and of primary renal proximal tubule epithelial cells (RPTEC) and assessed the relative mRNA abundance (normalized read counts) of each S1PR family gene (Figure 5D). These analyses revealed that different cell lines expressed different levels of the five genes, with *S1PR1* and *S1PR3* being the most abundantly expressed, arguing that these two receptors likely predominantly mediate S1P signaling in ccRCC cells. The inability of Sipo to inhibit S1PR_3_ signaling likely explains the lack of activity in soft agar assays in settings where FTY720 is effective and suggests that there is likely to be functional redundancy amongst the S1PR family proteins that are expressed in ccRCC cells. We next took advantage of a recently-published single-cell RNA-sequencing (scRNA-seq) dissection of eight human ccRCC tumors [37] to analyze patterns of S1PR gene family expression in different cell types. Highest expression levels were found in endothelial cells (*S1PR1*), fibroblasts (*S1PR2* and *S1PR3*) CD16^+^ monocytes, cycling CD8^+^ T-cells, effector T-helper cells (*S1PR4*), and FGFBP^+^ natural killer cells (*S1PR4* and *S1PR5*) (Figure 5E). We conclude that S1PR family genes are strongly and differentially expressed in several different cells of the tumor microenvironment including immune cells and stromal cells. This is in line with the well-known immunosuppressive effect of FTY720 and the vascular remodeling effect that has recently been described in ccRCC mouse models [19]. These observations highlight that sphingosine-based therapeutic approaches may have pleiotropic effects on many cells of the ccRCC tumor microenvironment. In contrast, while expression levels on tumor cells were lower than in other cells in the microenvironment, analyses of the three different tumor cell populations that were characterized by the original study (cycling tumor, tumor population 1, tumor population 2) revealed both diversity in terms of relative expression of the five genes between different tumor cell populations and also between different patient tumors (Figure 5F). Collectively, these observations argue that FTY720 that inhibits four of the S1PR family receptors is theoretically likely to be more therapeutically beneficial than Sipo that is restricted to inhibition of S1PR_1_ and S1PR_5_. 

### 3.6. HIF-α-Dependent and -Independent Effects of ACF and FTY720 

Since ACF and FTY720 treatments both inhibit the proliferation of cells irrespective of their sensitivity to HIF-2α inhibitors, and since both drugs are known to not only inhibit HIF-1α and HIF-2α, but also induce other effects such as DNA damage (ACF) or inhibit PI3K-mTOR pathway signaling (FTY720), we compared the effects of PT2399, ACF, Sipo, and FTY720 on the expression of HIF-2α target proteins (NDRG1 and CYCLIN D1) and on markers of activation of the PI3K/m-TOR pathway (phospho-Ser235/236 S6 ribosomal protein (P-S6) and phospho-Thr37/Thr46-4E-BP1 (P-4E-BP1)) in 786-O, 769-P, UMRC-2, and A498 cells (Figure 5G). As expected, 24 h of treatment with PT2399 reduced NDRG1 and CYCLIN D1 levels in all cell lines, albeit only weakly in UMRC-2 cells. FTY720 decreased CYCLIN D1 expression only in 769-P cells while ACF decreased CYCLIN D1 expression only in UMRC-2 cells. Neither FTY720 nor ACF inhibited NDRG1 expression in any of the cell lines. These observations suggest that while in some cells FTY720 and ACF can inhibit the expression of at least one important HIF-2α target gene, the anti-proliferative activities of these drugs are likely to be additional to, or independent of, effects on HIF-2α activity. Effects of the different drugs on readouts of mTORC1 activity also appear to uncouple this pathway from anti-proliferative effects of FTY720 and ACF. In 769-P cells, PT2399, FTY720, and ACF all reduced levels of P-S6 and P-4E-BP1, yet PT2399 does not inhibit proliferation of these cells. None of the drug treatments reduced P-S6 or P-4EBP1 levels in UMRC-2 cells, yet FTY720 and ACF inhibit soft agar growth of these cells. In A498 cells, Sipo reduced P-S6 and P-4E-BP1 levels but failed to exert any inhibitory effects on soft agar colony formation. We conclude that while FTY720 and ACF can both inhibit the expression of HIF-2α target genes in some settings, and can also inhibit mTORC1 activity in some settings, there are no clear correlations between these effects and the ability of these drugs to inhibit proliferation of ccRCC cells. Since S1PR family members can couple to several different G-proteins (G_i_, G_q_, G_12_, G_13_) and regulate numerous downstream signaling systems (AC, ERK, PLC, PI3K, Rho, JNK, ERK) [49] it is likely that the anti-proliferative effects of FTY720 in ccRCC cells are mediated by inhibition of multiple cellular signaling networks.

## 4. Conclusions

These studies collectively suggest that therapeutic approaches that aim to inhibit both HIF-1α and HIF-2α activities or that aim to inhibit S1P signaling should be further investigated in the clinical setting. Importantly these approaches may potentially show efficacy in the context of intrinsic or acquired resistance to HIF-2α inhibition, which is likely to become an important clinical issue.

## Figures and Tables

**Figure 1 cancers-13-04801-f001:**
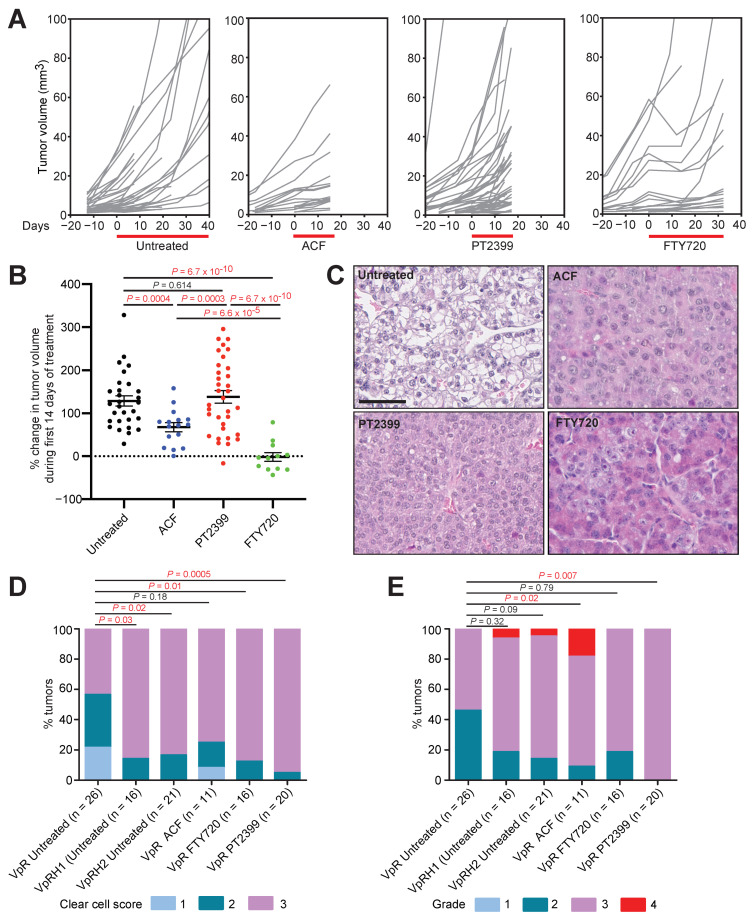
Therapy testing in the *Vhl/Trp53/Rb1* mutant mouse ccRCC model. (**A**) Individual tumor growth curves normalized to the day of assignment to experimental group and initiation of therapy (day 0). Treatment windows are depicted by the red bar. Curves are derived from 16, 3, 9, and 7 mice for the untreated, ACF, PT2399, and FTY720 cohorts, respectively. In some cases, multiple tumors were quantified per animal. (**B**) Maximum change in volume of individual tumors in the first 14 days of treatment. Mean ± SEM are derived from analyses of 29, 16, 34, and 12 tumors in untreated, ACF, PT2399, and FTY720 treated cohorts, respectively. *p*-values for pairwise comparisons were calculated by two-sided unpaired T-test with Welch’s correction. (**C**) Examples of histological appearances of untreated and treated tumors. Scale bar = 100 µm. (**D**,**E**) Scoring of clear cell phenotype (**D**) and nuclear grade (**E**) in the indicated numbers of *Vhl/Trp53/Rb1* (VpR) tumors treated with the indicated drugs. For comparison, previously published [15] scores of *Vhl/Trp53/Rb1/Hif1a* (VpRH1) and *Vhl/Trp53/Rb1/Hif2a* (VpRH2) mutant ccRCC tumors are included. *p*-values were calculated using the two-sided Mann–Whitney U test without adjustments for multiple comparisons.

**Figure 2 cancers-13-04801-f002:**
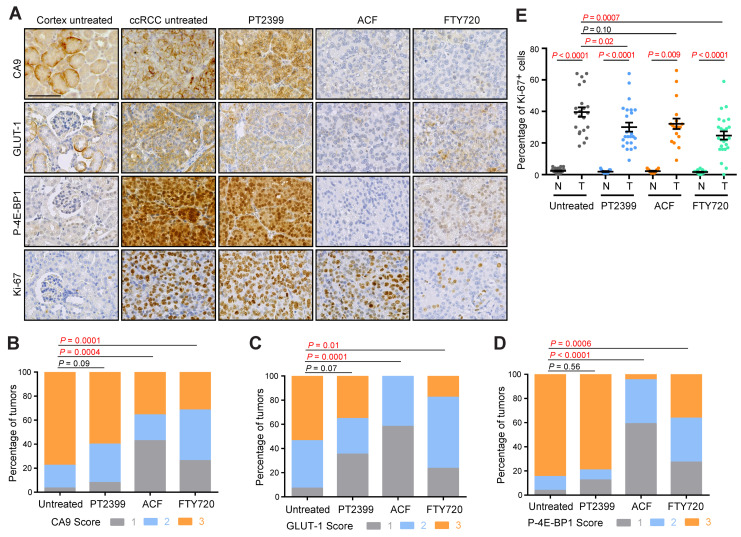
Molecular analyses of therapy testing in the *Vhl/Trp53/Rb1* mutant mouse ccRCC model. (**A**) Representative immunohistochemical stainings for CA9, GLUT-1, P-4E-BP1, and Ki-67 in regions of cortex and ccRCC from untreated mice and from regions of ccRCC from mice treated with PT2399, ACF, or FTY720. Scale bar = 100 mm. (**B**–**D**) Scoring of staining intensities of CA9 (**B**), GLUT-1 (**C**), and phospho-Thr37/Thr46-4E-BP1 (P-4E-BP1) (**D**) in ccRCC tumors. Analyses are based on 58, 25, 14, and 19 tumors for CA9, on 28, 17, 12, and 17 tumors for GLUT-1, and on 26, 24, 22, and 22 tumors for P-4E-BP1 in untreated, PT2399, ACF, and FTY720 treated cohorts. *p*-values were calculated using the two-sided Mann–Whitney U test without adjustments for multiple comparisons. Representative examples of scores 1, 2, and 3 for each staining are shown in Appendix A. (**E**) Quantification of percentage of Ki-67 positive nuclei in regions of normal cortex (N) and ccRCC (T). Analyses are based on 22, 24, 18, and 25 tumors in untreated, PT2399, ACF, and FTY720 treated cohorts. Mean ± SEM are shown, *p*-values for pairwise comparisons were calculated by Student’s *t*-test followed by two-sided Mann–Whitney U test without adjustments for multiple comparisons.

**Figure 3 cancers-13-04801-f003:**
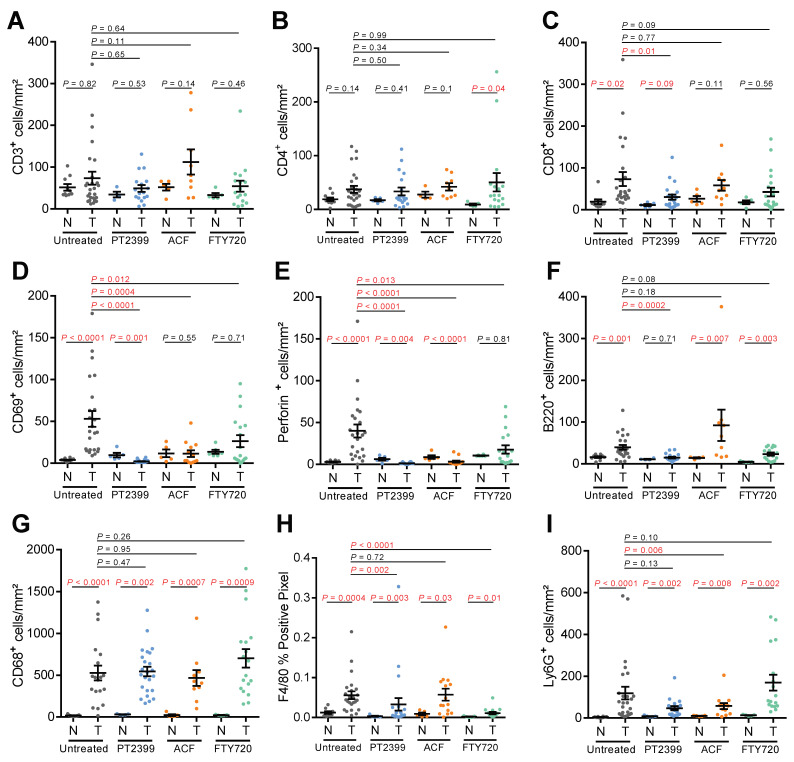
Analyses of the tumor immune microenvironment in therapy treated mice. (**A**–**I**) Quantification of the densities of immunohistochemically positive cells stained with the indicated antibodies in unaffected normal renal tissue (N) and VpR ccRCC tumors (T) in untreated (*n* = 19–26), PT2399 treated (*n* = 17–24), ACF-treated (*n* = 9–15), and FTY720-treated (*n* = 17–20) animals. Mean ± SEM are shown, *p*-values for pairwise comparisons were calculated by Student’s *t*-test followed by two-sided Mann–Whitney U test without adjustments for multiple comparisons.

**Figure 4 cancers-13-04801-f004:**
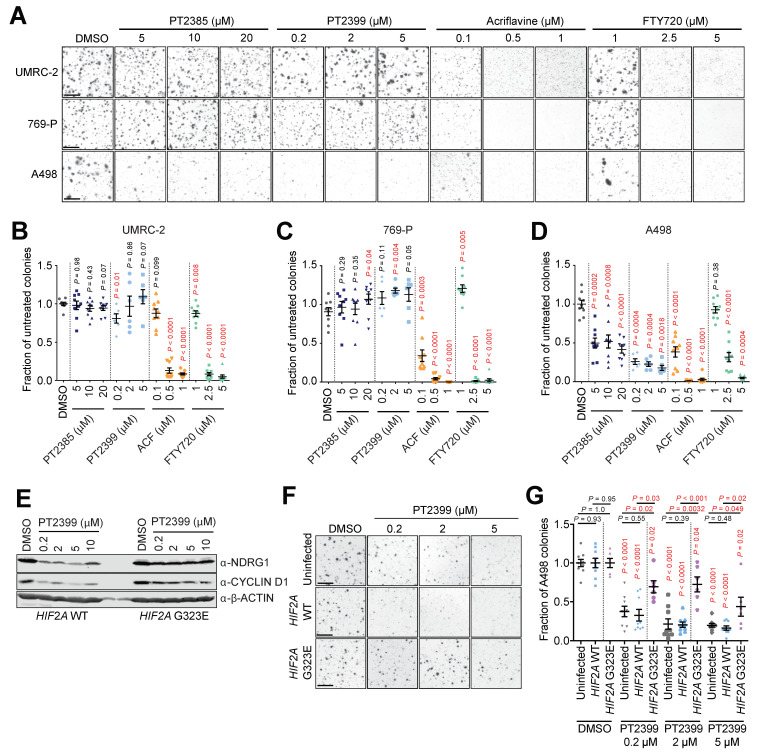
Inhibition of anchorage-independent growth of human ccRCC cells. (**A**) Representative images of stained soft agar colonies formed after 4 weeks of growth of UMRC-2, 769-P, and A498 cells in the presence of DMSO (control) or the indicated doses of the indicated drugs. Scale bar = 100 μm. (**B**–**D**) Quantifications of colony formation, expressed as the fraction of colonies formed in untreated cultures in UMRC-2 (**B**), 769-P (**C**), and A498 (**D**) cells treated with the indicated doses of the indicated drugs. Data are derived from 2 (for PT2399) and 3 (all other treatments) independent experiments, each conducted in triplicate and are displayed as mean ± SEM. *p*-values for pairwise comparisons were calculated by Student’s t-test followed by two-sided Mann–Whitney U test without adjustments for multiple comparisons. (**E**) Immunoblots of *HIF2A* WT (left panel) and *HIF2A* G323E (right panel) for NDRG1, CYCLIND D1, and β-ACTIN after 48 h of treatment with DMSO or PT2399 in the indicated concentrations. (**F**) Representative images of stained soft agar colonies formed after 4 weeks of growth of A498, *HIF2A* WT, and *HIF2A* G323E cells in presence of DMSO (control) or the indicated doses of PT2399. Scale bar = 100 µm. (**G**) Quantifications of colony formation, expressed as the fraction of colonies formed in cultures of A498, *HIF2A* WT, and *HIF2A* G323E cells treated with DMSO or PT2399 in the indicated concentrations. Data are derived from 2 (*HIF2A* G323E) and 3 (A498 and *HIF2A* WT) independent experiments, each conducted in triplicate and are displayed as Mean ± SEM. *p*-values for pairwise comparisons were calculated by Student’s *t*-test followed by two-sided Mann–Whitney U test without adjustments for multiple comparisons.

**Figure 5 cancers-13-04801-f005:**
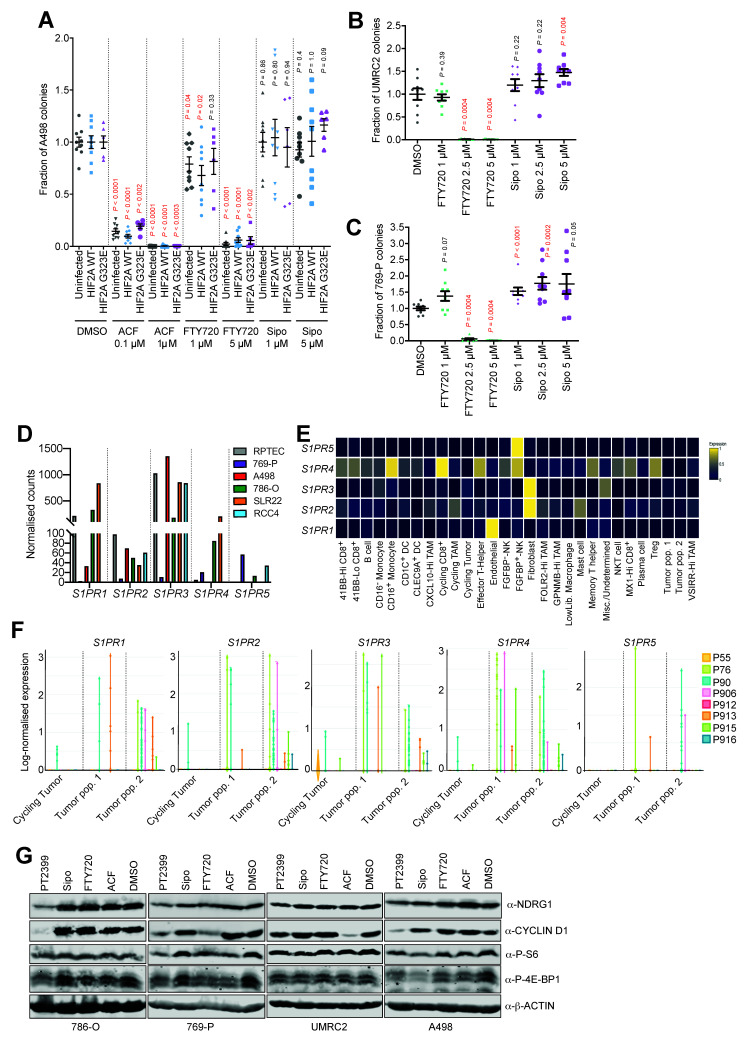
Multiple S1P receptors contribute to ccRCC tumor cell proliferation. (**A**–**C**) Quantifications of colony formation, expressed as the fraction of colonies formed in cultures of A498, *HIF2A* WT, *HIF2A* G323E, UMRC-2, and 769-P cells treated with DMSO or the indicated drugs in the indicated concentrations. Data are derived from 2 (*HIF2A* G323E) and 3 (all other cell lines) independent experiments, each conducted in triplicate and are displayed as mean ± SEM. *p*-values for pairwise comparisons were calculated by Student’s *t*-test followed by two-sided Mann–Whitney U test without adjustments for multiple comparisons. (**D**) Relative mRNA expression of S1PR_1–5_ from RNA-sequencing of the indicated human ccRCC cell lines and of primary renal proximal tubule epithelial cells (RPTEC). (**E**) Sc-RNA-seq based heatmap (blue: low expression, yellow: high expression) showing cell specific expression of S1PR_1–5_ derived from eight human ccRCC samples. (**F**) Log-normalized expression of S1PR_1–5_ of three defined ccRCC populations derived from sc-RNA-seq of eight human ccRCC samples; labeling of individual patient tumors (P55 etc.) is as defined in the original publication [37]. (**G**) Immunoblots of indicated ccRCC cell lines for NDRG1, CYCLIN D1, P-S6, P-4-EBP1, and β-ACTIN after 24 h of treatment with DMSO, PT2399 (10 µM), Sipo (10 µM), FTY720 (10 µM), or ACF (2.5 µM).

## Data Availability

Source data are provided with this paper. All remaining relevant data are available in the article, Appendix A, or from the corresponding author upon reasonable request.

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
