# Peer review of "Therapeutic Effects of Inhibition of Sphingosine-1-Phosphate Signaling in HIF-2α Inhibitor-Resistant Clear Cell Renal Cell Carcinoma"

_cancers, 2021, doi:10.3390/cancers13194801_

Round 1
Reviewer 1 Report
This resubmitted manuscript by Hoefflin et al. has thoroughly addressed all the comments and concerns from the reviewer. In addition, new experimental evidence has been added in this new version. I feel that the quality and scientific soundness of the manuscript have been largely improved. The data is adequate and interesting to the researchers studying ccRCC from the therapeutic aspect.
Reviewer 2 Report
Authors stated that “ABCA1 and SPNS2 but not ABCC1 were highly expressed in the tumors (Supplementary Figure S3) [line 301]. However, in Fig S3, SPNS2 and ABCC1 seem to be overexpressed, but ABCC1 does not. Please double check your statement.
This manuscript is a resubmission of an earlier submission. The following is a list of the peer review reports and author responses from that submission.
Round 1
Reviewer 1 Report
This manuscript investigated the effects of three direct and indirect HIF-α inhibitors on tumor growth and change in the tumor microenvironment in the ccRCC mouse model. Some of the data especially the in vivo and in vitro antitumor effect of these drugs is interesting. Although the data are well presented, some of the data could not fully support the conclusion and the topic defined by the title. In general, there are several concerns from the reviewer as listed below.
- It's unnecessary to show the SFMBT1 downstream genes. Their upregulation (except SPHK1) is uncorrelated with the SIP pathway activation. In addition, only 4 genes were showed upregulated in mouse tumors vs. cortex, 6 genes were upregulated in human ccRCC samples. furtherly, only two common genes (SPHK1 and ABCA1) upregulated in both data sets. According to these data, it doesn't suggest the sphingosine pathway is activated in ccRCC.
- While multiple tumors were measured in each mouse. Indicate how many mice are contained in each treatment cohort. Why there are some tumors were stopped monitoring in the untreated group as shown in Figure 1A? How were these tumors considered in the quantification as shown in Figure 1B?
- The relationship between HIF signaling and the sphingosine pathway is unknown and should be described in the introduction. What's the rationale to use these inhibitors or targeting these pathways in this study should be clarified. From the abstract FTY720 is an inhibitor targeting the S1P signaling pathway, while from Line 224-225, authors declaimed that it inhibits HIF-1/2a activity. Again, the mechanism of action of this inhibitor should be introduced more.
- From Figures 1C and D, PT2399 caused significant clear cell phenotype reduction while didn't inhibit tumor growth. How to explain the inconsistent tumor inhibition and clear cell phenotype reduction caused by these drugs?
- From the results of the current drug treatment, which contains two HIF1/2a co-inhibition and one HIF-2a inhibition, it's hard to conclude that the tumor growth inhibition is independent of HIF-2a. HIF-1a and HIF-2a have a considerable amount of common downstream target genes. Inhibit either one may not sufficiently block the oncogenic signaling pathways regulated by both.
- The effect of the tumor immune microenvironment could be discussed more. A thorough discussion part is necessary for this study.
Reviewer 2 Report
Authors described the efficacy of HIF-1alpha/2alpha inhibitor and S1P signaling inhibitor. They generated the mouse model with kidney epithelium-specific deletion of the Vhl, Trp53 and Rb1, which mimics ccRCC phenotypes, in the previous study. Using this model, they examined the effect of potential therapeutic drugs in tumor growth, grade, tumor immune microenvironment. Finally, their effects were tested in HIF-2alpha-resistant ccRCC cell lines.
Although this study is worthy presenting the future therapeutic direction, there is not enough evidence about their efficacy in HIF-2alpha-resistant RCC.
Major comments
1. Authors showed that target genes of SFMBT transcription factor and S1P pathway were increased in ccRCC than normal tissues in Suppl Fig. 1. Authors need to check whether these drugs are specifically effective through your interesting pathways and whether the tumor indicators (size, molecular signaling, immune environment) are truly correlated with these signaling. After treatment of three drugs, the target genes of S1P signaling and HIF signaling should be confirmed in these tissues. Then, it should be examined whether the inhibitory pattern of these expression levels is correlated with tumor regression.
2. Authors needs to confirm whether Acriflavin and FTY720 have inhibitory effects in HIF-2alpha-resistant A498 cells. As you described, you can use sgRNA-mediated HIF-2alpha-KO A498 cells and then check the effect of acriflavine and FTY720 in Fig. 4.
Minor comments
1. Please double check whether authors aim to explain like this “Interestingly, ACF and PT2399 treatments, but not FTY720 treatment, shifted the distribution towards higher nuclear grade (Figure 1E).
2. Please unify the representative name of Fingolimod or FTY720.